# Impacts of body composition parameters and liver cirrhosis on the severity of alcoholic acute pancreatitis

**Dong Kee Jang[1]ᵒ, Dong-Won Ahn[1]ᵒ, Kook Lae Lee[1], Byeong Gwan Kim[1], Ji Won Kim[1], Su Hwan Kim[1], Hyoun Woo Kang[1], Dong Seok Lee[1], Soon Ho Yoon[2], Sang Joon Park[2], Ji Bong Jeong [1]***

**1** Department of Internal Medicine, Seoul National University College of Medicine, Seoul Metropolitan Government Seoul National University Boramae Medical Center, Seoul, Republic of Korea, **2** Department of Radiology, Seoul National University College of Medicine, Seoul National University Hospital, Seoul, Republic of Korea

ᵒ These authors contributed equally to this work.

\* jibjeong@gmail.com

**Data Availability Statement:** All relevant data are within the manuscript and its Supporting Information files.

## Abstract

### Aim

Liver cirrhosis and features of muscle or adipose tissues may affect the severity of acute pancreatitis (AP). We aimed to evaluate the impact of body composition parameters and liver cirrhosis on the severity of AP in patients with alcohol-induced AP (AAP).

### Methods

Patients with presumed AAP who underwent CT within one week after admission were retrospectively enrolled. L3 sectional areas of abdominal fat and muscle, and mean muscle attenuations (MMAs) were quantified. The presence of liver cirrhosis was determined using clinical and CT findings. Factors potentially associated with moderately severe or severe AP were included in the multivariable logistic regression analysis.

### Results

A total of 242 patients (47.0 ± 12.6 years, 215 males) with presumed AAP were included. The mild and moderately severe/severe (MSS) groups included 137 (56.6%) and 105 patients (43.4%), respectively. Patients in the MSS group had higher rates of liver cirrhosis, organ failure, and local complications. Among body composition parameters, mean MMA (33.4 vs 36.8 HU, $P<0.0001$) and abdominal muscle mass (126.5 vs 135.1 cm$^2$, $P = 0.029$) were significantly lower in the MSS group. The presence of liver cirrhosis (OR, 4.192; 95% CI, 1.620–10.848) was found to be a significant risk factor for moderately severe or severe AP by multivariable analysis.

### Conclusion

The results of this study suggest that liver cirrhosis has a significant impact on the severity of AAP. Of the body composition parameters examined, MMA and abdominal muscle mass showed potential as promising predictors.

**Funding:** The authors received no specific funding for this work.

**Competing interests:** NO authors have competing interests.

## Introduction

In the United States, acute pancreatitis (AP) is one of the leading gastrointestinal causes of hospitalization [1], and the incidences of both alcohol-induced and biliary AP are increasing [2]. Alcohol is responsible for about 30% of AP cases in the US [3], and alcoholic liver disease is the leading cause of death among non-malignant gastrointestinal diseases [1]. Chronic alcohol use places individuals at high risk of AP and alcoholic liver diseases, and thus, factors that determine the severities of alcohol-induced AP (AAP) and liver disease are expected to be similar.

Fifteen to twenty percent of patients with AP progress to severe or complicated AP [4, 5]. Since the mortality rates of mild and severe AP are quite different ($< 5\%$ [6] and 36–50%, respectively) [7–9], evaluating AP severity in patients with early AP is important to determine treatment strategies and the need to transfer to an intensive care unit or an advanced facility. After the revised Atlanta classification system was issued, moderately severe and severe AP have been defined according to organ failure and local complications [10]. Several scoring systems, such as the Ranson [11], APACHE-II [12], and BISAP [13] systems have been developed, but they are not widely used in practice because they are either outdated or complex. Researchers have tried to predict AP severity using various laboratory parameters such as C-reactive protein (CRP), blood urea nitrogen (BUN), and procalcitonin [14–16], but it has proven to be difficult to predict severity accurately using a single indicator.

Several recent studies have shown that computed tomography (CT) defined muscle and adipose tissue features are associated with AP severity and mortality [17–24]. Since most patients undergo CT during the diagnosis of AP, predictions of prognosis of AP by simple body composition analyses using CT images is a cost-effective proposition. Subcutaneous adipose tissue (SAT) are [18–22], visceral adipose tissue (VAT) area [17–24], skeletal muscle mass or density [17–21, 24], visceral fat-to-muscle ratio (VMR) [21, 22], and mean muscle attenuation (MMA) [19, 21] have all been evaluated in the context of CT-defined body composition analysis. In addition, fatty liver has also been reported to affect AP severity [25]. However, the results of studies vary and indicate the impacts of fatty liver and body composition parameters depends on ethnicity and geographical region.

Patients with AP show various clinical features that depend on etiology. For example, Choi et al. reported that a higher body mass index (BMI) increases the risks of gallstone and non-gallstone-related AP but has a greater impact on the risk of gallstone-related AP. Furthermore, sex, alcohol intake, and smoking have been reported to be associated with the risk of AP in the low BMI range for non-gallstone-related AP [26]. However, most studies have been conducted on all AP patients regardless of etiology, and thus, results are not homogeneous. Furthermore, the body composition parameters evaluated in previous studies differed. Therefore, in the present study, we limited subjects to AAP patients and evaluated relationships between AP severity and body composition parameters or liver cirrhosis in these patients.

## Methods

### Study subjects

The medical and radiographic records of adult patients with a diagnosis of first-time AP hospitalized at the Seoul Metropolitan Government Seoul National University Boramae Medical Center from 2011 to 2020 were identified in this retrospective study. Only subjects with presumed AAP and a history of sufficient alcohol intake were included in the present study. All those with a high probability of AP due to other causes such as chronic pancreatitis, post-endoscopic retrograde cholangiopancreatography pancreatitis, periampullary cancer, intraductal

papillary mucinous neoplasm, and autoimmune pancreatitis were excluded. Subjects that did not undergo CT within one week of admission were also excluded. A diagnosis of AP was established if at least two of the following three features were present: (1) acute upper-abdominal pain; (2) a serum lipase or amylase level at least three times greater than the upper limit of normal; and (3) characteristic contrast-enhanced CT findings of AP [10]. The included AAP patients were divided into two groups according to AP severity (mild and moderately severe/severe). The study protocol was approved beforehand by our Institutional Review Board (IRB No. 30-2020-308), which waived the requirement for informed consent.

## Definitions

AP severity was determined using the revised Atlanta classification [10], according to which mild AP has neither organ failure nor local or systemic complications, moderately severe AP is defined by the presence of transient organ failure or local complications, and severe AP is defined by persistent organ failure ($>$ 48 h) according to the modified Marshall scoring system [27]. The presence of local complications, including acute peripancreatic fluid collection and acute necrotic collection, was also determined using the revised Atlanta classification [10]. Clinical diagnoses of liver cirrhosis were made as follows: (1) a platelet count of $<$ 100,000/μL and CT findings compatible with cirrhosis (a blunted, nodular liver surface, and splenomegaly); or (2) clinical signs of portal hypertension (ascites, varix, or hepatic encephalopathy) [28]. Child-Pugh class was determined as previously described [29]. Fatty liver was defined as a liver-to-spleen attenuation ratio of $<$ 1 in unenhanced CT images [25].

## CT-based body composition analysis

Earliest abdominal CT scans obtained within 1 week of hospitalization were analyzed. All abdominal CT scans were performed using a 64-slice multidetector CT scanner (Brilliance 64; Philips Healthcare, Amsterdam, The Netherlands). Abdominal CT images were uploaded to commercially available deep learning-based software for body composition analysis (Deep-Catch[R] v1.0.0.0; Medicalip Co. Ltd., Seoul). This software package provides automatic volumetric segmentation of the following seven body components with an accuracy of 97% [30]: skin, muscle, abdominal visceral fat, subcutaneous fat, bone, internal organs and vessels, and the central nervous system. In addition, the software provides automatic localization of the third lumbar vertebral body (L3), and automatically quantifies L3 sectional area ($cm^2$) and mean CT attenuations (Hounsfield Units, HU) of visceral abdominal fat, subcutaneous abdominal fat, and abdominal muscle components (Fig 1). One radiologist (S.H.Y.) with 16 years of experience of body CT interpretation unaware of clinical information confirmed the appropriateness of the automatic segmentations of body components. The MMAs were measured by averaging CT attenuation of the abdominal muscles at the L3 level, including abdominal wall muscles, psoas muscles, quadratus lumborum muscles, and paraspinal muscles. The MMAs decrease as the amount of intramuscular fat, myosteatosis, increases [31].

## Statistical analysis

Categorical variables are presented as numbers and percentages, and continuous variables as means ± standard deviations. For univariable analysis, categorical variables were compared using Pearson's chi-squared test or Fisher's exact test, and continuous variables were compared using Student's t-test or the Mann–Whitney's U test, as appropriate. Variables with $P$ values of $<$ 0.10 by univariable analysis and other variables considered necessary to control their contributions were entered into the multivariable logistic regression analysis. All analyses

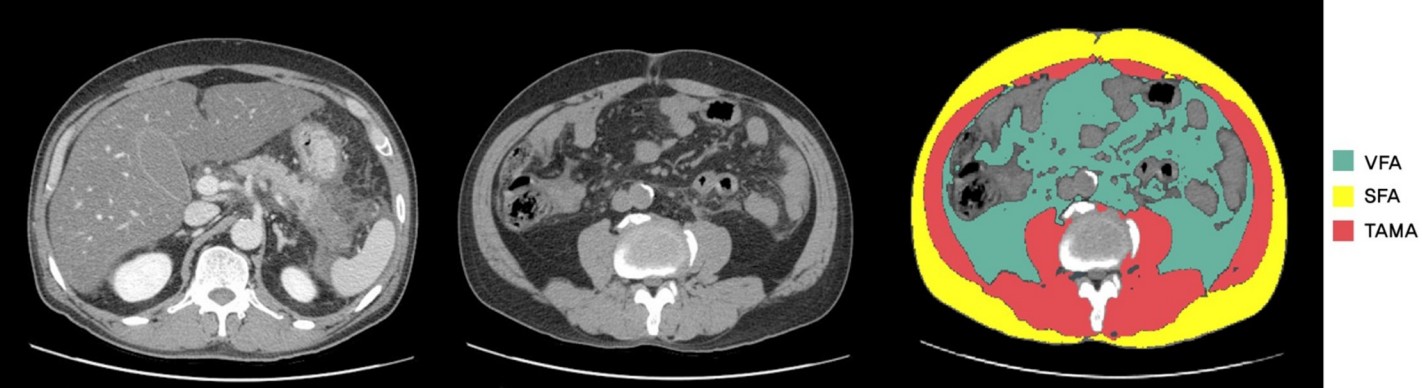

**Fig 1. Computed tomography (CT) findings and body morphometric evaluations of abdominal fat and muscle areas in an acute pancreatitis patient (M/56) who had consumed more than 47 g daily.** (A). A portal phase CT image showing necrosis, edematous change, and peripancreatic fluid collection in the pancreatic tail. (B/C) Noncontrast and mapped CT images obtained using DeepCatch$^R$ at the same level of the inferior endplate of the L3 vertebra. (C) Segmented axial CT image showing visceral fat area (VFA, cm$^2$), subcutaneous fat area (SFA, cm$^2$), and total abdominal muscle area (TAMA, cm$^2$) for psoas, paraspinals, transversus abdominis, rectus abdominis, quadratus lumborum, and internal and external obliques.

were performed using SPSS version 24.0 (IBM Corp., Armonk, New York, USA) and *P* values of < 0.05 were considered statistically significant.

## Results

### Baseline characteristics and body composition parameters

A total of 242 patients (mean age, 47.0 ± 12.6 years; 215 males, 88.8%) with presumed AAP were enrolled and divided into two groups (a mild group and a moderately severe or severe (MSS) group), which contained 137 (56.6%) and 105 patients (43.4%), respectively. The process of selecting subjects with presumed AAP is described in Fig 2. Table 1 summarizes the baseline characteristics of patients in the two study groups and comorbidities, BMI, BUN,

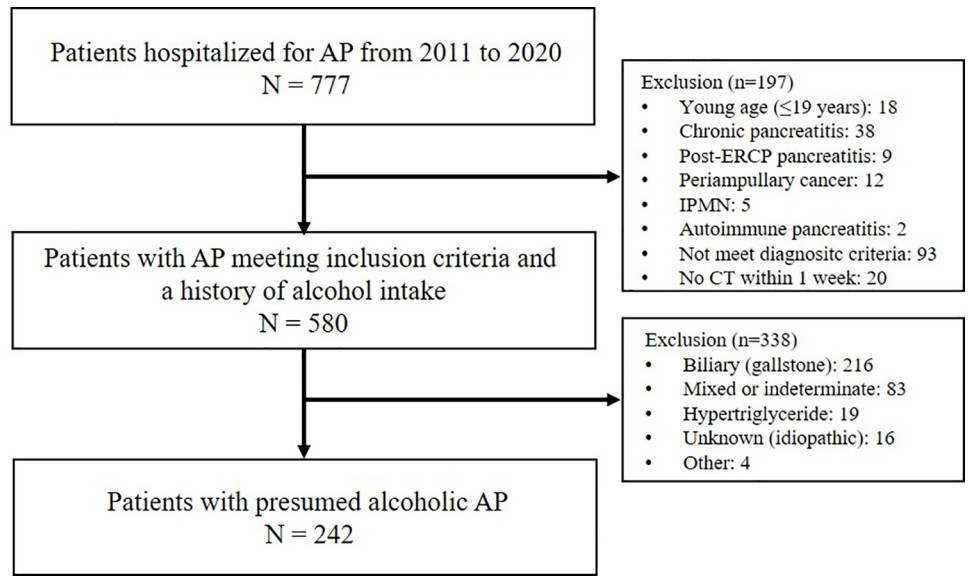

**Fig 2. Flow-chart of patient selection.** AP, acute pancreatitis; ERCP, endoscopic retrograde cholangiopancreatography; IPMN, intraductal papillary mucinous neoplasm; CT, computed tomography.

**Table 1. Patient baseline characteristics (N = 242).**

| Variables | Total | Mild | Moderately severe/severe | P value |
|---|---|---|---|---|
| | (N = 242) | (N = 137, 56.6%) | (N = 105, 43.4%) | |
| Age (y) | 47.0 ± 12.6 | 45.1 ± 12.6 | 49.4 ± 12.3 | 0.007 |
| Male, n (%) | 215 (88.8%) | 124 (90.5%) | 91 (86.7%) | 0.346 |
| Comorbidities, n (%) | | | | |
| Liver cirrhosis | 32 (13.2%) | 7 (5.1%) | 25 (23.8%) | <0.001 |
| Diabetes | 38 (15.7%) | 17 (12.4%) | 21 (20.0%) | 0.108 |
| Hypertension | 69 (28.5%) | 34 (24.8%) | 35 (33.3%) | 0.146 |
| Hypercholesterolemia | 27 (11.2%) | 16 (11.7%) | 11 (10.5%) | 0.768 |
| BMI (kg/m$^2$) | 21.9 ± 16.3 | 23.3 ± 21.0 | 20.0 ± 6.0 | 0.087 |
| BUN (mg/dL) | 16.1 ± 13.7 | 14.1 ± 6.3 | 18.7 ± 19.3 | 0.020 |
| Organ failure, n (%) | 63 (26.0%) | 0 (0%) | 63 (60.0%) | <0.001 |
| Local complication, n (%) | 66 (27.3%) | 0 (0%) | 66 (62.9%) | <0.001 |
| APFC | 31 (12.8%) | 0 (0%) | 31 (29.5%) | |
| ANC | 35 (14.5%) | 0 (0%) | 35 (33.3%) | |
| Hospital days (day) | 9.5 ± 14.8 | 6.4 ± 4.4 | 13.6 ± 21.3 | 0.001 |

BMI, body mass index; BUN, blood urea nitrogen; APFC, acute peripancreatic fluid collection; ANC, acute necrotic collection.

organ failure, local complications, and hospital days. Patients in the MSS group were significantly older, had a higher mean BUN level, and higher rates of liver cirrhosis. Mean hospital days were also greater in the MSS group (13.6 vs 6.4 days; 95% CI, 3.0–11.4; $P = 0.001$). However, mean BMI tended to be lower in the MSS group (20.0 vs 23.3 kg/m$^2$; 95% CI, -7.2–0.5; $P = 0.087$). Table 2 provides a comparison of body composition parameters in the two groups. MMA (33.4 vs 36.8 HU; 95% CI, -5.1 –-1.5; $P < 0.0001$) and abdominal muscle mass (126.5 vs 135.1 cm$^2$; 95% CI, -16.4 - -0.9; $P = 0.029$) were significantly lower in the MSS group, while VAT areas, SAT areas, VMRs, and fatty liver rates were not significantly different.

## Risk factors for moderately severe or severe alcoholic AP

Univariable and multivariable results for risk factors of moderately severe or severe AAP are presented in Table 3. The adjusted odds ratio (OR) of the presence of liver cirrhosis was highest with significance (OR 4.192; 95% CI, 1.620–10.848). BUN ≥ 20 mg/dL was also a significant risk factor (OR 2.845; 95% CI, 1.320–6.133). Among body composition parameters, an MMA of < 36 HU was significantly associated with moderately severe or severe AAP in the

**Table 2. Body composition parameters and severities of alcoholic acute pancreatitis.**

| Variables | Total | Mild | Moderately severe/severe | P value |
|---|---|---|---|---|
| | (N = 242) | (N = 137, 56.6%) | (N = 105, 43.4%) | |
| Visceral adipose tissue area (cm$^2$) | 123.5 ± 67.3 | 123.9 ± 67.7 | 122.8 ± 67.1 | 0.900 |
| Subcutaneous adipose tissue area (cm$^2$) | 117.5 ± 64.6 | 119.7 ± 70.6 | 114.7 ± 56.1 | 0.556 |
| Abdominal muscle mass (cm$^2$) | 131.4 ± 30.5 | 135.1 ± 32.0 | 126.5 ± 27.9 | 0.029 |
| Visceral fat-to-muscle ratio | 0.93 ± 0.43 | 0.90 ± 0.42 | 0.96 ± 0.44 | 0.335 |
| Mean muscle attenuation (HU) | 35.3 ± 7.2 | 36.8 ± 6.8 | 33.4 ± 7.4 | <0.001 |
| Fatty liver | 155 (64.0%) | 83 (60.6%) | 72 (68.6%) | 0.165 |

HU, Hounsfield Units.

**Table 3. Risk factors of moderately severe or severe alcoholic acute pancreatitis.**

| Factor | Univariable OR (*P* value, 95% CI) | Multivariable OR (*P* value, 95% CI) |
|---|---|---|
| Age (y) | 1.028* | 1.010 |
| | (0.008, 1.007–1.050) | (0.462, 0.984–1.037) |
| BMI < 22 kg/m$^2$ | 1.306 | 1.224 |
| | (0.392, 0.709–2.408) | (0.597, 0.579–2.585) |
| BUN ≥ 20 mg/dL | 3.168* | 2.845* |
| | (0.001, 1.623–6.185) | (0.008, 1.320–6.133) |
| Liver cirrhosis | 5.804* | 4.192* |
| | (<0.001, 2.400–14.036) | (0.003, 1.620–10.848) |
| Abdominal muscle mass < 127 cm$^{2\dagger}$ | 1.493 | 0.785 |
| | (0.124, 0.896–2.488) | (0.509, 0.382–1.611) |
| Mean muscle attenuation < 36 HU$^\dagger$ | 2.610* | 1.722 |
| | (<0.001, 1.547–4.403) | (0.110, 0.884–3.352) |
| Fatty liver | 1.464 | 1.298 |
| | (0.166, 0.854–2.510) | (0.417, 0.692–2.435) |

*$P$ value < 0.05

$^\dagger$Dichotomized at the median level.

Hosmer-Lemeshow goodness of fit test (multiple logistic regression analysis), *P* value = 0.1574.

OR, odds ratio; BMI, body mass index; BUN, blood urea nitrogen.

univariable analysis (OR 2.610, 95% CI, 1.547–4.403). Characteristics of cirrhotic patients (N = 32) are described in Table 4. Clinical and body composition factors were not significantly different in the two groups. In particular, the difference in AAP severity according to Child-Pugh class was not significant. However, abdominal muscle mass was inversely correlated with AP severity in patients with liver cirrhosis and this correlation was greater for liver cirrhosis patients in the MSS group (115.9 vs 99.9; 95% CI, 4.5–27.3; *P* = 0.008).

**Table 4. Characteristics of cirrhotic patients with alcoholic acute pancreatitis (N = 32).**

| Variables | Mild (N = 7, 21.9%) | Moderately severe/severe (N = 25, 78.1%) | P value |
|---|---|---|---|
| Age (y) | 45.1 ± 13.3 | 53.4 ± 9.6 | 0.075 |
| Male, n (%) | 4 (57.1%) | 23 (92.0%) | 0.057 |
| BMI (kg/m$^2$) | 32.2 ± 44.5 | 19.3 ± 7.0 | 0.474 |
| BUN (mg/dL) | 12.4 ± 3.6 | 20.4 ± 16.5 | 0.032 |
| Visceral adipose tissue area (cm$^2$) | 89.8 ± 25.3 | 94.3 ± 66.4 | 0.784 |
| Subcutaneous adipose tissue area (cm$^2$) | 89.0 ± 44.2 | 87.9 ± 55.9 | 0.963 |
| Abdominal muscle mass (cm$^2$) | 99.9 ± 8.4 | 115.9 ± 22.8 | 0.008 |
| Visceral fat-muscle ratio | 0.90 ± 0.24 | 0.80 ± 0.46 | 0.570 |
| Mean muscle attenuation (HU) | 33.2 ± 2.6 | 31.8 ± 7.2 | 0.436 |
| Hospital days (day) | 9.3 ± 9.0 | 17.3 ± 34.6 | 0.308 |
| Child-Pugh class | | | 0.327 |
| A | 3 (42.9%) | 5 (20.0%) | |
| B or C | 4 (57.1%) | 20 (80.0%) | |

BMI, body mass index; BUN, blood urea nitrogen.

## Discussion

In the current study, we undertook to evaluate the impacts of body composition parameters and liver cirrhosis on AP severity in patients with AAP. We hypothesized that cirrhosis, sarcopenia, and the distribution of adipose tissue in the abdominal cavity would significantly impact the severity of AAP. We found liver cirrhosis had a greater effect on the severity of AAP than body composition parameters. Among body composition parameters, only MMA < 36 HU significantly impacted AAP severity by univariable analysis (OR, 2.610; 95% CI, 1.547–4.403). These results suggest that the outcomes of AAP patients substantially depend on underlying liver function, which contrasts with the effects of other etiologies such as gallstone pancreatitis. However, to date, few studies have addressed the effects of cirrhosis or body composition in AAP.

A previous meta-analysis reported that obese individuals, especially those with a BMI > 30 kg/m$^2$, developed significantly more severe AP (summary relative risk, 1.82; 95% CI, 1.44–2.30) than nonobese patients [32]. However, in our study, the severity of AP was rather high in patients with a low BMI, which might be expected given the poor nutritional statuses of chronic alcohol drinkers [33]. Furthermore, the average BMI of patients included in this study was only 21.9 kg/m$^2$, which prevented comparisons with the results of previous studies that used a BMI cut-off of 30 kg/m$^2$. In the present study, we used a BMI cut-off of 22 kg/m$^2$ in our risk factor analysis. In terms of laboratory variables, a BUN level of $\geq$ 20 mg/dL significantly predicted AP severity, which agrees with the result of a previous international validation study [34]. CRP level was excluded from the analysis because it is difficult to accurately reflect the severity of AP in the early stage [35, 36]. In addition, creatinine and creatinine clearance levels were also excluded because they were used for the definition of organ failure needed in severity classification, and procalcitonin was excluded because there were too many missing values.

Furthermore, the effect of sarcopenia as determined by CT also conflicted with previous reports. Trikudanathan et al. recently reported that decreased skeletal muscle density independently predicted mortality in necrotizing pancreatitis [17], whereas Sternby et al. showed that lower muscle mass levels were associated with less severe AP [21]. These conflicting results may have been due to different patient compositions. We found lower abdominal muscle mass was associated with more severe AP by univariable analysis (126.5 vs 135.1 cm$^2$; 95% CI, -16.4 - -0.9; $P$ = 0.029). Taken together, it appears that impacts of BMI and sarcopenia on AP patients are dependent on patient characteristics. We suggest further research be conducted to define sarcopenia based on abdominal muscle mass determined by CT.

A lower MMA level, indicating myosteatosis, showed a potential association with AP severity in the present study, which concurs with a report by Sternby et al [21]. However, no cut-off value has been determined to define myosteatosis on abdominal CT images, and thus, we dichotomized MMA values about the median level (36 HU), while Sternby et al. [21] classified their patient population using tertiles (median value, 29.3 HU). Low muscle attenuation (low intramuscular fat content) has previously been related to poor prognosis and increased risk of complications and morbidity, especially in cancer patients, independently of BMI [37]. Myosteatosis indicates possible malnutrition [38], which might have an adverse effect on the prognosis of AP, but further studies are needed to provide a more information on this topic.

The reported impacts of visceral and subcutaneous adipose tissue areas on AP severity differ widely. Yoon et al. reported that both VAT area and VMR were strongly correlated with AP severity, and suggested that a VMR of 1 is an optimal threshold for predicting moderately severe or severe pancreatitis [22]. Natu et al. also concluded that increased VAT area strongly predicts severe AP [23]. However, Shimonov et al. showed that higher amounts of visceral fat were positively associated with lower recurrence [20], which is somewhat similar to our results

although the outcome variables used were different. Low adipose tissue area is thought to indicate malnutrition in patients with AAP, which is characteristic of chronic alcoholics, and thus, previous results would be expected to differ depending on how many chronic alcohol drinkers were included. Our study included many homeless chronic drinkers due to the nature of our hospital, as was reflected by the very low mean BMI of the included patients.

In our study, liver cirrhosis was found to best predict AAP severity (adjusted OR, 4.192; 95% CI, 1.620–10.848) and to do so independently of fatty liver. Recently, two retrospective cohort studies on the clinical outcomes of AP in patients with cirrhosis were published [39, 40]. A single-center study concluded that morbidity and mortality among AP patients with or without cirrhosis were similar [39], whereas a national cohort study conducted in the USA reported AP patients with cirrhosis had higher inpatient mortality (OR 3.4, $P < 0.001$) [40]. However, AP severity was not assessed in the US national cohort study, and only 19 alcoholic cirrhosis patients were included in the single-center study [39]. Furthermore, AP patients with all etiologies were included in both studies. In contrast, we only included AAP patients, and in most liver cirrhosis was alcohol-induced. The fact that cirrhosis was found to well predict AP severity in our study probably stems from peripheral arterial vasodilation caused by systemic inflammation leading to organ failure [41]. Furthermore, concomitant alcoholic ketoacidosis or acute hepatitis may have affected outcomes. However, there was no significant difference in the AAP severity according to Child-Pugh class in our result, probably because there were small number of cirrhotic patients included.

The current study has several limitations that warrant consideration. First, it is inherently limited by its retrospective, single-center design, which makes it somewhat difficult to generalize our results, though we suggest they can be better understood as characteristics of chronic alcoholics with AAP. Second, the timing of CT was not unified, and local complications may have been dependent on CT timing. Third, we did not assess amounts of alcohol consumed, as most of the patients included were chronic drinkers, it was difficult to obtain accurate details of medical history at admission. Despite these limitations, we conducted our analysis on a homogeneous group of patients with AAP and excluded all patients with ambiguous causes, and thus, we believe our results are meaningful and identify some important characteristics of AAP patients.

In conclusion, our results indicate that liver cirrhosis has a significant impact on the severity of AAP. Among body composition parameters, MMA and abdominal muscle mass showed potential as promising predictors, but we suggest further studies be conducted to properly define myosteatosis based on CT determined MMA levels and that future studies consider cirrhosis and accompanying diseases, such as acute hepatitis or ketoacidosis, to improve understanding of AAP. A large-scale prospective study is also needed to better understand the relationship between AAP severity and body composition parameters or liver cirrhosis.

## Supporting information

**S1 File. Data file.**
(XLSX)

## Acknowledgments

The authors would like to thank Pf. Sohee Oh, a statistician who provided statistical advice.

## Author Contributions

**Conceptualization:** Ji Bong Jeong.

**Data curation:** Dong Kee Jang, Dong-Won Ahn, Kook Lae Lee, Ji Bong Jeong.

**Formal analysis:** Dong Kee Jang, Dong-Won Ahn, Byeong Gwan Kim.

**Investigation:** Dong Kee Jang, Dong-Won Ahn, Ji Won Kim, Su Hwan Kim, Soon Ho Yoon, Sang Joon Park.

**Methodology:** Hyoun Woo Kang, Dong Seok Lee, Soon Ho Yoon, Sang Joon Park.

**Supervision:** Ji Bong Jeong.

**Writing – original draft:** Dong Kee Jang, Dong-Won Ahn.

**Writing – review & editing:** Dong Kee Jang, Dong-Won Ahn, Ji Bong Jeong.

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
