## [Decision Letter · Decision Letter 0]

6 Sep 2021

PONE-D-21-22616Impacts of body composition parameters and liver cirrhosis on the severity of alcoholic acute pancreatitisPLOS ONE

Dear Dr. Jeong,

Thank you for submitting your manuscript to PLOS ONE. After careful consideration, we feel that it has merit but does not fully meet PLOS ONE’s publication criteria as it currently stands. Therefore, we invite you to submit a revised version of the manuscript that addresses the points raised during the review process.

Your paper has been evaluated by two experts in the field (Reviewers #1 and #2) as well as a statistician (Reviewer #3). All have found merit in your study but think that the pateints should be better characterized (e.g. how many had signs of chronic pancreatitis,  whether only patients with the first episode of acute pancreatitis were involved in the analysis or whether patients had mild and moderate/severe acute pancreatitis)

We look forward to receiving your revised manuscript.

Kind regards,

Zoltán Rakonczay Jr., M.D., Ph.D., D.Sc.

Academic Editor

PLOS ONE

Journal Requirements:

“NO”

Reviewers' comments:

Reviewer's Responses to Questions

**Comments to the Author**

1. Is the manuscript technically sound, and do the data support the conclusions?

Reviewer #1: Yes

Reviewer #2: Partly

Reviewer #3: Partly

2. Has the statistical analysis been performed appropriately and rigorously? 

Reviewer #1: Yes

Reviewer #2: Yes

Reviewer #3: Yes

3. Have the authors made all data underlying the findings in their manuscript fully available?

Reviewer #1: Yes

Reviewer #2: Yes

Reviewer #3: Yes

4. Is the manuscript presented in an intelligible fashion and written in standard English?

Reviewer #1: Yes

Reviewer #2: Yes

Reviewer #3: Yes

5. Review Comments to the Author

Reviewer #1: In this retrospective study Jang et al. analyzed the effect of body composition, i.e. abdominal fat and muscle and mean muscle attenuation, and concurrent liver cirrhosis in patients with acute alcoholic pancreatitis. These patients were divided into a group with mild acute pancreatitis and moderate or severe acute pancreatitis (MSS). The authors found a higher rate of liver cirrhosis and a decreased muscle mass and muscle attenuation in patients with moderate and severe acute pancreatitis.

The study covers an interesting topic since risk factors/associated factors for severe acute pancreatitis have not entirely investigated yet. The association between myosteatosis, measured by mean muscle attenuation in CT, and severity of acute pancreatitis is interesting and seems to indicate underlying malnutrition, as the authors also discussed in their manuscript.

There are some points that need to be further clarified.

Major comments:

1.) The methods sections needs to be extended and it should be described how mean muscle attenuation was detected and to what extent it indicates myosteatosis.

2.) It would be interesting to know, how many patients also had signs of chronic pancreatitis because chronic alcohol consumption not only predisposes to liver cirrhosis but also to chronic pancreatitis.

3.) What was the Child-Pugh Score for liver cirrhosis in the patient group of mild vs. moderate/severe acute pancreatitis? It can be assumed that individuals with a lower muscle mass and moderate/severe acute pancreatitis more likely suffered from advanced stages of liver cirrhosis. These data should be incorporated into the manuscript.

Minor comment:

1.) The authors should also mention the subdivision of patients in mild and moderate/severe acute pancreatitis in their methods section.

Reviewer #2: This study investigates the effect of body composition and liver cirrhosis on the severity of acute alcoholic pancreatitis. The authors concluded that liver cirrhosis has a negative impact on the outcome of acute alcoholic pancreatitis, furthermore, lower mean muscle attenuation is a risk factor for moderately severe and severe pancreatitis.

This is a retrospective, single center study involving 242 patients from a 10-year long period.

It not known whether only patients with the first episode of acute pancreatitis were involved in the analysis. It is important since repeated episodes of acute alcoholic pancreatitis are quite common. The repeated episodes are causing irreversible changes in the pancreas, local complications occurs more frequently, and the pancreatitis can be more severe. This is a potential bias, therefore should be considered and patients with repeated attacks should be analyzed separately.

Line 58: outdated

Line 59: use laboratory parameters instead of hematological indicators

Line 138: organ failure and local complications are defining the severity of pancreatitis, therefore it is obvious that they are occurring more frequently in the MSS group. These data are also shown in table one, the comparison of mild and MSS groups does not make much sense.

Reviewer #3: This is a secondary data analysis manuscript, trying to evaluate the impact of body composition parameters and liver cirrhosis on the severity of AP in Korean patients with alcohol-induced AP (AAP). The study was approved by the respective ethics board. The writeup looks straightforward, and the statistical analysis methods were described clearly. I do have the following comments:

(a) Can the authors provide a sample size/power paragraph, based on what effect sizes was envisioned (particularly, in the multiple logistic regression analysis)? The study recruited 242 patients with presumed AAP; however, some context on sample size/power would provide the reader a basis to understand how many subjects should be recruited in future studies of similar kinds. The sample size should be calculated, using the primary endpoint.

(b) The multiple logistic regressions should be followed by a desired goodness-of-fit assessment, say, via the Hosmer-Lemeshow statistic. This is missing in this submission.

(c) The authors should make sure that explanation of covariate effects in the Results section should always be followed with the estimate, and respective 95% interval.

(d) The Discussion should allude to future studies conducted on a larger scale on subjects recruited at other countries to better understand the relationships between AP severity, and body composition parameters or liver

cirrhosis.

6. PLOS authors have the option to publish the peer review history of their article (what does this mean?). If published, this will include your full peer review and any attached files.

Reviewer #1: No

Reviewer #2: No

Reviewer #3: No

---

## [Author Response · Author response to Decision Letter 0]

13 Oct 2021

Please refer to the attached file, "response to reviewer's comments_PONE-D-21-22616".

---

## [Decision Letter · Decision Letter 1]

8 Nov 2021

Impacts of body composition parameters and liver cirrhosis on the severity of alcoholic acute pancreatitis

PONE-D-21-22616R1

Dear Dr. Jeong,

We’re pleased to inform you that your manuscript has been judged scientifically suitable for publication and will be formally accepted for publication once it meets all outstanding technical requirements.

Kind regards,

Zoltán Rakonczay Jr., M.D., Ph.D., D.Sc.

Academic Editor

PLOS ONE

Additional Editor Comments (optional):

Reviewers' comments:

Reviewer's Responses to Questions

**Comments to the Author**

1. If the authors have adequately addressed your comments raised in a previous round of review and you feel that this manuscript is now acceptable for publication, you may indicate that here to bypass the “Comments to the Author” section, enter your conflict of interest statement in the “Confidential to Editor” section, and submit your "Accept" recommendation.

Reviewer #1: All comments have been addressed

Reviewer #3: All comments have been addressed

2. Is the manuscript technically sound, and do the data support the conclusions?

Reviewer #1: Yes

Reviewer #3: (No Response)

3. Has the statistical analysis been performed appropriately and rigorously? 

Reviewer #1: Yes

Reviewer #3: (No Response)

4. Have the authors made all data underlying the findings in their manuscript fully available?

Reviewer #1: Yes

Reviewer #3: (No Response)

5. Is the manuscript presented in an intelligible fashion and written in standard English?

Reviewer #1: Yes

Reviewer #3: (No Response)

6. Review Comments to the Author

Reviewer #1: The authors have sufficiently answered the reviewers‘ queries and the manuscript became more clear now.

Reviewer #3: (No Response)

7. PLOS authors have the option to publish the peer review history of their article (what does this mean?). If published, this will include your full peer review and any attached files.

Reviewer #1: No

Reviewer #3: No

---

## [Editor Report · Acceptance letter]

11 Nov 2021

PONE-D-21-22616R1 

Impacts of body composition parameters and liver cirrhosis on the severity of alcoholic acute pancreatitis 

Dear Dr. Jeong:

I'm pleased to inform you that your manuscript has been deemed suitable for publication in PLOS ONE. Congratulations! Your manuscript is now with our production department. 

Kind regards, 

on behalf of

Dr. Zoltán Rakonczay Jr. 

Academic Editor

PLOS ONE